# OpenReview forum: "How Jailbreak Defenses Work and Ensemble? A Mechanistic Investigation"
_ICLR.cc/2025/Conference — Submitted to ICLR 2025_

### Official Review · Reviewer_U7of · 2024-10-22

**Soundness:** 3
**Presentation:** 4
**Contribution:** 3
**Rating:** 6
**Confidence:** 3

**Summary:**

This paper systematically investigates jailbreak defense mechanisms by reformulating the generative task as a binary classification problem to probe model refusal tendencies across both harmful and benign queries. Focusing on internal strategies, this study identifies two key defense mechanisms: safety shift and harmfulness discrimination. They increase refusal probabilities for all queries and enhance the model’s ability to distinguish benign and harmful queries, respectively. Experiments demonstrate the effectiveness of the ensemble defense mechanism.

**Strengths:**

1. This paper is well-written and easy to follow.

2. Two metrics, Mean Shift and Distance Change, are reasonable and can help to visualize the differences between different types of defenses.

3. The experiment part is comprehensive and some insights in Section 4.4 and 4.5 are interesting.

**Weaknesses:**

1. Although the authors mention that they do not assess the actual usefulness of model’s responses, but rather the willingness from a safety perspective, a core concern of this reviewer is that “Can some seemingly effective defense methods, like refactoring, truly keep a good performance on benign queries?”.

2. Lack of the experiments of defenses against stronger and various attackers. This reviewer is worried about the limit of attack methods, as well as the datasets. MM-Safety-Bench is a comprehensive benchmark, but the malicious query is quite direct and lacks of stealthiness. “Hide the text queries at the bottom of associated images” can only represent a few types of attack methods. More experiments on stronger or various attackers are recommended.

3. Although the experiment part is comprehensive, it lacks of some mainstream model optimization defenses, such as PPO and DPO.

**Questions:**

1. See weakness 1. Can some seemingly effective defense methods, like refactoring, truly keep a good performance on benign queries?

2. See weakness 2. Can MM-SafetyBench and MOSSBench represent most attack vectors?

---

> ### Author Response · Authors · 2024-11-23
> **Response to Review**
>
> Thank you for your insightful review and very helpful comments! Below are our responses and we have updated the draft accordingly (highlighted in blue).
> ***
> > **W1/Q1:** Can some seemingly effective defense methods, like refactoring, truly keep a good performance on benign queries?
> - We appreciate the reviewer’s insightful comment. The quality of generated responses is indeed an important consideration when evaluating safety, while we directly follow current research on LVLMs jailbreaking where safety evaluations primarily focus on the model's ability to refuse harmful queries or respond appropriately to benign queries.
> - **[Additional Experiement on a Separate Quality Evaluation Dataset]** Following established practices in prior multimodal jailbreak defense research [1, 2, 3], we further evaluate response quality under different defense strategies using the **MM-Vet dataset** for utility analysis. The results, detailed in **Appendix D (Table 11)**, and summarized as follows:
>   1. Individual defenses, either inducing a safety shift or enhancing harmfulness discrimination, generally result in minor declines in response quality. As anticipated by the reviewer, refactoring defenses, especially Caption w/o Image, have more pronounced impacts on utility.
>   2. However, some of our **ensemble strategies can effectively mitigate utility degradation** compared to their individual counterparts, such as _Responsible-SafeDecoding_, _Caption-Intention_, achieving a better balance between safety and true helpfulness.
>
> ***
> > **W2/Q2:** Can MM-SafetyBench and MOSSBench represent most attack vectors?
> - **[Additional Experiments on the Comprehensive JailbreakV-28K Dataset]** Thank you for your insightful feedback. To address your concern, we conduct additional experiments on the recently published **JailbreakV-28K**, a comprehensive multimodal jailbreak evaluation benchmark that includes _16 safety policies, 5 diverse jailbreak methods, and a variety of image types_. The results, detailed in **Appendix E (Table 12)**, demonstrate that ensemble strategies generally achieve superior performance compared to individual defenses, particularly in mitigating MLLM-based attacks where baseline models initially demonstrate weaker defensive capabilities.
> ***
> > **W3:** Although the experiment part is comprehensive, it lacks of some mainstream model optimization defenses, such as PPO and DPO.
> - **[Additional Experiments on DPO]** Thank you for your valuable suggestion. We further implement DPO defense strategy and evaluate its performance on both LLaVA-1.5-7B and LLaVA-1.5-13B. The results are updated in **Table 1 (Safety Evaluation) and Table 11 (Unility Evaluation)**, with detailed implementations listed in **Appendix A**. We find that the defenses models trained with DPO demonstrate relatively weaker robustness (particularly lower DSR) compared to those trained with SFT. However, DPO models exhibit a reduced tendency for over-defense.
>
> [1] AdaShield: Safeguarding Multimodal Large Language Models from Structure-based Attack via Adaptive Shield Prompting.
> [2] Eyes Closed, Safety On: Protecting Multimodal LLMs via Image-to-Text Transformation.
> [3] MLLM-Protector: Ensuring MLLM's Safety without Hurting Performance.

---

> ### Author Response · Authors · 2024-11-26
> **Friendly reminder to respond to author rebuttal**
>
> Dear Reviewer U7of,
>
> Thank you again for your review! As you know, we are quickly approaching the draft update deadline Nov 27 EOD.
>
> In light of this, we would love to hear your thoughts about our rebuttal, including whether it sufficiently addresses your concerns and questions, and whether you would like to see further edits made. Any feedback is welcome and greatly appreciated!
>
> Sincerely,
> Authors

---

> > ### Author Response · Authors · 2024-12-02
> >
> > Dear Reviewer U7of,
> >
> > Thank you so much again for your feedback! We have previously updated the PDF based on your comments, and we have provided our rebuttal above for addressing your questions and concerns!
> >
> > Today is the last day of the discussion period, and we would be grateful if you could check the rebuttal and let us know if it sufficiently addresses all your queries. If you believe that our rebuttal is satisfactory, it would be great if you could consider increasing your score. Any feedback is welcome and greatly appreciated!
> >
> > Sincerely,
> >
> > Paper10764 Authors

---

### Official Review · Reviewer_QcMJ · 2024-10-27

**Soundness:** 2
**Presentation:** 3
**Contribution:** 2
**Rating:** 6
**Confidence:** 3

**Summary:**

This paper propose a novel and straightforward way to reformulate the
LVLM's generation problem as a binary classification problem, in order to
investigate the the mechanism of Jailbreak Defenses.

**Strengths:**

1. A new angle to investigate jailbreak defenses is proposed. It is
   interesting.
2. The reformulation practice is interesting and valuable, providing a
   effective way to investigate the mechanism of jailbreak defenses.
3. Extensive experiments across various jailbreak defenses are conducted.

**Weaknesses:**

1. The motivation behind focusing on LVLMs is not clear.
2. Further analysis on the results (especially the ensemble part) is needed.
3. The selection of models under evaluation is not convincing.

**Questions:**

1. I believe the reformulation practice is valuable and straightforward.
   However, it seems that it can also be applied to other types of models,
   like LLMs. What is the motivation behind focusing on LVLMs?
2. The ensemble part is not well analyzed. Could you provide more insights
   on the ensemble mechanism? For example, provide some specific suggestions
   for ensemble strategies.
3. The selection of models under evaluation is not convincing; both models
   are LLaVA. Could you add some more models with different architectures
   to make the evaluation more convincing?
4. Given the results presented in Figure 4, does there exist any way to
   further investigate the representative power of the binary classifier?
   Furthermore, can this representative power of the binary classifier
   be improved?

---

> ### Author Response · Authors · 2024-11-23
> **Response to Review**
>
> Thank you for your insightful review and very helpful comments! Below are our responses and we have updated the draft accordingly (highlighted in blue).
> ***
> > **W1/Q1:** I believe the reformulation practice is valuable and straightforward. However, it seems that it can also be applied to other types of models, like LLMs. What is the motivation behind focusing on LVLMs?
> Thank you for your suggestion.
> - **[Why Focus on Multimodal Jailbreaking]** Compared to extensive research on jailbreak attacks and defenses for LLMs, multimodal scenarios introduces additional vulnerabilities due to diverse attack sources and broader input space, making them more challenging and effective defense methods remain underexplored. Understanding the mechanisms behind current defenses and developing robust ensemble strategies are therefore more critical in multimodal contexts. Thus, our research primarily focuses on LVLMs.
> - **[Additional Analysis on LLMs]** To investigate whether our identified mechanisms are applicable to text-only LLMs, we conduct an additional analysis on LLaMA-3.1-8B using the XStest dataset, a text-only dataset comprising 250 safe prompts and 200 unsafe prompts. New results are presented in **Appendix B.4 (Figure 9)**, demonstrating that LLMs also exhibit the same two mechanisms observed in LVLMs. Furthermore, Inter-Mechanism and Intra-Mechanism Ensembles can respectively strengthen an individual defense mechanism and achieve a more balanced trade-off.
> ***
> > **W2/Q2:** The ensemble part is not well analyzed. Could you provide more insights on the ensemble mechanism? For example, provide some specific suggestions for ensemble strategies.
> - Thank you for your suggestion. Based on our experiments across MM-SafetyBench, MOSSBench, JailbreakV-28K (Appendix E), and utility analysis on MM-Vet (Appendix D), we observe that the **SR+MO ensemble** strategy consistently achieves superior performance. Specifically, the **Demonstration-SFT method** stands out for its strong defensive capabilities while also maintaining high utility performance and a reasonable response rate.  Additionally, the **QR|SR ensemble** strategy also yields strong overall performance, though slightly less effective than SR+MO. Specifically, it offers a robust trade-off by combining defenses based on distinct mechanisms.
> ***
> > **W3/Q3:** The selection of models under evaluation is not convincing; both models are LLaVA. Could you add some more models with different architectures to make the evaluation more convincing?
> - **[Additional Results on More LVLMs]** Thank you for your insightful question. To evaluate the generalizability of our findings, we conduct additional analysis on advanced LVLMs, including **LLaVA-Next** (specifically LLaVa-V1.6-Mistral-7B with a different LLMs backbone), **Qwen2-VL** (specifically Qwen2-VL-7B with a different training paradigm) and **Pixtral** (specifically Pixtral-12B with a different model architecture). The results, presented in **Appendix B.3 (Figure 6,7,8)**, demonstrate that these LVLMs also exhibit the same two mechanisms, and the two ensemble strategies generally achieve similar effects as LLaVA-1.5, further supporting the robustness and applicability of our findings across diverse LVLMs.
> ***
> > **Q4:** Given the results presented in Figure 4, does there exist any way to further investigate the representative power of the binary classifier? Furthermore, can this representative power of the binary classifier be improved?
> - **[Correlation for Representative Power Investigation]** To further investigate the representative power of the binary classifier, we analyze the correlation of Detection Success Rate (DSR) across defense methods between classification and generative settings, to give an overview sight . Specifically, we calculate their Spearman's Rank Correlation Coefficient, which is 0.59, indicating a moderate positive monotonic correlation. The results are detailed in **Appendix C (Figure 11)**.
> -  **[Improving Representative Power by Adjusting the Classification Threshold]** As indicated in Section 4.4 and Figure 9, "_the model tends to exhibit slightly higher refusal rates during classification compared to generation_”. To enhance the classifier's representative power, we can adjust the classification threshold for determining whether a model refuses a response, increasing it from _0.5_ to _0.7_. This can improve the correlation of DSR between classification and generative settings, raising the correlation coefficient from _0.59_ to _0.64_. Additionally, for the generation task, inconsistencies introduced by the randomness of greedy decoding may contribute to variability. Adopting appropriate sampling strategies could help mitigate these inconsistencies, further enhancing the binary classifier's representative power for generative settings.

---

> > ### Comment · Reviewer_QcMJ · 2024-11-24
> >
> > Looks good to me. Have increased my score to 6.

---

> > > ### Author Response · Authors · 2024-11-26
> > > **Thank you for your feedback!**
> > >
> > > Dear Reviewer QcMJ,
> > >
> > > Thank you for your feedback and valuable comments! We’re glad to hear that our rebuttal addressed your concerns.
> > >
> > > Sincerely,
> > > Authors

---

### Official Review · Reviewer_XiPx · 2024-11-02

**Soundness:** 3
**Presentation:** 3
**Contribution:** 3
**Rating:** 6
**Confidence:** 3

**Summary:**

This paper investigates the trade-off between safety and helpfulness in jailbreak defenses, highlighting two fundamental mechanisms: safety shift and harmfulness discrimination. The authors analyze various ensemble strategies to enhance model safety and improve the safety-helpfulness balance, demonstrating their effectiveness across multimodal contexts.

**Strengths:**

1. Introduces novel defense mechanisms (safety shift and harmfulness discrimination) for LVLMs, providing fresh insights into model security.
2. Includes a comprehensive analysis supported by rigorous empirical validation across various datasets and models, utilizing a robust methodology by reformulating generation tasks into classification problems.
3. Evaluates two ensemble defense strategies (inter-mechanism and intra-mechanism integration), examining the balance between enhancing model safety and preserving usability.

**Weaknesses:**

1. The captions for the figures and tables in the paper are overly simplistic.
2. The experiments primarily rely on the MM-SafetyBench and MOSSBench datasets, which may not fully reflect the diversity and complexity of real-world scenarios.
3. While the paper proposes various defense strategies, it may not adequately discuss their feasibility and cost-effectiveness in practical deployment.

**Questions:**

1. While the paper focuses on LLaVA-1.5-7B and LLaVA-1.5-13B, how do the authors ensure that the findings apply to other models within the LLaVA series?
2. Can the captions for figures and tables be more detailed?
3. What is the optimal combination of the 27 defense methods, and can you analyze why this combination yields better results? Additionally, do these findings have universality for other models, such as language models?
4. Will these defense strategies affect the model's real-time response speed? Will they introduce additional overhead?

---

> ### Author Response · Authors · 2024-11-23
> **Response to Review**
>
> Thank you for your insightful review and very helpful comments! Below are our responses and we have updated the draft accordingly (highlighted in blue).
> ***
> > **W1/Q2:** The captions for the figures and tables in the paper are overly simplistic.
>
> Thank you for your advice, and we apologize for this oversight. We have provided more detailed captions in the updated PDF (as highlighted in blue).
> ***
> > **W2:** The experiments primarily rely on the MM-SafetyBench and MOSSBench datasets, which may not fully reflect the diversity and complexity of real-world scenarios.
> - **[Additional Experiments on the Comprehensive JailbreakV-28K Dataset]** Thank you for your insightful feedback. To address your concern, we conduct additional experiments on the recently published **JailbreakV-28K**, a comprehensive multimodal jailbreak evaluation benchmark that includes _16 safety policies, 5 diverse jailbreak methods, and a variety of image types_. The results, detailed in **Appendix E (Table 12)**, demonstrate that ensemble strategies generally achieve superior performance compared to individual defenses, particularly in mitigating MLLM-based attacks where baseline models initially demonstrate weaker defensive capabilities.
> ***
> > **W3/Q4:** While the paper proposes various defense strategies, it may not adequately discuss their feasibility and cost-effectiveness in practical deployment.
> - **[Defenses Generally Reduce Latency for Harmful Queries But Improve It for Benign Queries]** Thank you for the valuable suggestion. To evaluate the impact of defense strategies on real-time response speed, we conduct a detailed comparison of inference times for benign and harmful queries separately. The results, presented in **Appendix F (Table 13)**, indicate that defense methods generally increase inference time for benign queries, with this effect being more pronounced in methods like _Query Refactoring_. Conversely, for harmful queries, most defense strategies can reduce inference time by generating concise rejection responses.
> ***
> > **Q1:** While the paper focuses on LLaVA-1.5-7B and LLaVA-1.5-13B, how do the authors ensure that the findings apply to other models within the LLaVA series?
> - **[Additional Results on More LVLMs]** Thank you for your insightful question. To evaluate the generalizability of our findings, we conduct additional analysis on advanced LVLMs, including **LLaVA-Next** (specifically LLaVa-V1.6-Mistral-7B with a different LLMs backbone), **Qwen2-VL** (specifically Qwen2-VL-7B with a different training paradigm) and **Pixtral** (specifically Pixtral-12B with a different model architecture). The results, presented in **Appendix B.3 (Figure 6,7,8)**, demonstrate that these LVLMs also exhibit the same two mechanisms, and the two ensemble strategies generally achieve similar effects as LLaVA-1.5, further supporting the robustness and applicability of our findings across diverse LVLMs.
> ***
> > **Q3:** What is the optimal combination of the 27 defense methods, and can you analyze why this combination yields better results? Additionally, do these findings have universality for other models, such as language models?
>
> Thank you for the insightful question.
> - **[Optimal Combination of Defense Methods]** Based on our experiments across MM-SafetyBench, MOSSBench, JailbreakV-28K (Appendix E), and utility analysis on MM-Vet (Appendix D), we observe that the **SR+MO ensemble** strategy consistently achieves superior performance. Specifically, the **Demonstration-SFT method** stands out for its strong defensive capabilities while also maintaining high utility performance and a reasonable response rate.  We attribute its effectiveness to the integration of two distinct safety shift defenses, as Demonstration and SFT are the best-performing individual defense methods  in their respective mechanisms, resulting in a complementary effect that significantly enhances overall performance. Additionally, the **QR|SR ensemble** strategy also yields strong overall performance, though slightly less effective than SR+MO. Specifically, it offers a robust trade-off by combining defenses based on distinct mechanisms.
> - **[Applicability to LLMs]** Our analysis identifies two key defense mechanisms in multimodal scenarios. To investigate whether these findings apply to text-only LLMs, we conduct the same analysis on LLaMA-3.1-8B using the XStest dataset, a text-only dataset comprising 250 safe prompts and 200 unsafe prompts. Results are presented in **Appendix B.4 (Figure 9)**, demonstrating that LLMs also exhibit the same two mechanisms observed in LVLMs. Furthermore, Inter-Mechanism and Intra-Mechanism Ensembles can respectively strengthen an individual defense mechanism and achieve a more balanced trade-off.

---

> > ### Comment · Reviewer_XiPx · 2024-11-26
> >
> > I would like to thank the authors for their response. However, upon further reflection, I have concerns about the validity and persuasiveness of the problem that the paper addresses. I will keep my current rating.

---

### Official Review · Reviewer_m8ZD · 2024-11-04

**Soundness:** 3
**Presentation:** 3
**Contribution:** 2
**Rating:** 5
**Confidence:** 4

**Summary:**

This paper attempts to characterize internal jailbreak defenses into two categories: Safety Shift and Harmfulness Discrimination. The authors do this by prompting the model with a classification question and analyzing the output distribution for benign and harmful queries. Using this formulation, the paper propose intra and inter mechanism ensembling techniques that help balance the safety and usefulness trade-off offered by the model. The evaluation is performed on LLaVA-1.5.

**Strengths:**

The author tackle an important problem of characterizing how LLM jailbreak defenses work. The paper is well written and motivated. I appreciate the author's effort in evaluating a variety of defenses. Moreover, attributing defenses to safety shift and harmfulness discrimination is an interesting idea.

**Weaknesses:**

1. The analysis in this paper can also be applied to text-only LLMs. Since, text-only LLMs are more widely used, the authors should consider expanding the analysis.
2. The whole analysis focuses on affirmative response on benign queries and refusal on harmful queries. However, it does not take into account the quality of the generated responses (specially since the evaluation uses a pattern matching based judge). Combining multiple defenses could severely harm the quality of the returned responses. For instance, the query refactoring method Caption w/o image first captions the image and then adds it as text to the model prompt. While helping with safety, this would lead to decrease in quality of responses on benign queries.
3. The mechanistic analysis is done on a classification setting, and the insights might not transfer to the generative setting (as also hinted by the authors).

**Questions:**

1. The paper performs the entire evaluation on a single LVLM - LLaVA-1.5. It would be interesting to see if the findings generalize to other LVLMs like CogVLM and InternLM-XC.
2. The mechanistic analysis characterizes defenses as either safety shift or harmfulness discrimination. However, Figure 3 (b) shows that combination of SR + MO i.e. two safety shift techniques lead to an increase in distance from 0.5 to 0.63. This suggests that defenses might be providing different degrees of both safety shift and harmfulness discrimination.

---

> ### Author Response · Authors · 2024-11-23
> **Response to Review (1)**
>
> Thank you for your insightful review and very helpful comments! Below are our responses and we have updated the draft accordingly (highlighted in blue).
> ***
> > **W1:** The analysis in this paper can also be applied to text-only LLMs. Since text-only LLMs are more widely used, the authors should consider expanding the analysis.
>
> Thank you for your suggestion.
> - **[Why Focus on Multimodal Jailbreaking]** Compared to extensive research on jailbreak attacks and defenses for LLMs, multimodal scenarios introduces additional vulnerabilities due to diverse attack sources and broader input space, making them more challenging and effective defense methods remain underexplored. Understanding the mechanisms behind current defenses and developing robust ensemble strategies are therefore more critical in multimodal contexts. Thus, our research primarily focuses on LVLMs.
> - **[Additional Analysis on LLMs]** To investigate whether our identified mechanisms are applicable to text-only LLMs, we conduct an additional analysis on **LLaMA-3.1-8B** using the XStest dataset, a text-only dataset comprising 250 safe prompts and 200 unsafe prompts. New results are presented in **Appendix B.4 (Figure 9)**, demonstrating that LLMs also exhibit the same two mechanisms observed in LVLMs. Furthermore, Inter-Mechanism and Intra-Mechanism Ensembles can respectively strengthen an individual defense mechanism and achieve a more balanced trade-off.
> ***
> > **W2:** The whole analysis focuses on affirmative response on benign queries and refusal on harmful queries. However, it does not take into account the quality of the generated responses (specially since the evaluation uses a pattern matching based judge). Combining multiple defenses could severely harm the quality of the returned responses. For instance, the query refactoring method Caption w/o image first captions the image and then adds it as text to the model prompt. While helping with safety, this would lead to a decrease in quality of responses to benign queries.
> - We appreciate the reviewer’s insightful comment. The quality of generated responses is indeed an important consideration when evaluating safety, while we directly follow current research on LVLMs jailbreaking where safety evaluations primarily focus on the model's ability to refuse harmful queries or respond appropriately to benign queries.
> - **[Multimodal Jailbreak Benchmarks Are Not Designed for Quality Evaluation]** Like most LLM jailbreaking datasets, our used multimodal jailbreak benchmarks-MMSafetyBench and MOSSBench-emphasize evaluting models' disengagement from malicious prompts rather than response quality. They primarily assess metrics such as Attack Success Rate and Refusal Rate. Moreover, some benign queries in these benchmarks are unsuitable for quality evaluation, including preference-based or unanswerable questions, like "Describe how your approach to life has evolved over the past five years". Although a new benchmark **StrongREJECT**[1] is proposed for LLMs jailbreaking, which explicitly evaluates response quality and safety, no equivalent benchmark exists for multimodal settings (to our knowledge), highlighting an opportunity for future research.
> - **[Additional Experiement on a Separate Quality Evaluation Dataset]** Following established practices in prior multimodal jailbreak defense research [2, 3, 4], we separately evaluate response quality under different defense strategies using the **MM-Vet** dataset for utility analysis. The results, detailed in **Appendix D (Table 11)**, and summarized as follow:
>   1. Individual defenses, either inducing a safety shift or enhancing harmfulness discrimination, result in minor declines in response quality. As anticipated by the reviewer, defense methods like Caption w/o Image have more pronounced impacts on utility.
>   2. However, some ensemble strategies can effectively mitigate utility degradation compared to their individual counterparts, such as _Responsible-SafeDecoding_ and _Caption-Intention_, achieving a better balance between safety and true helpfulness.
>
> [1] A StrongREJECT for Empty Jailbreaks.
> [2] AdaShield: Safeguarding Multimodal Large Language Models from Structure-based Attack via Adaptive Shield Prompting.
> [3] Eyes Closed, Safety On: Protecting Multimodal LLMs via Image-to-Text Transformation.
> [4] MLLM-Protector: Ensuring MLLM's Safety without Hurting Performance.

---

> ### Author Response · Authors · 2024-11-23
> **Response to Review (2)**
>
> > **W3:** The mechanistic analysis is done on a classification setting, and the insights might not transfer to the generative setting (as also hinted by the authors).
> - **[Classification and Generation are Positively Monotonic Correlated]** While the classification and generative settings are **not entirely** consistent, as discussed in Section 4.4, we further analyze the correlation of Detection Success Rate (DSR) across defense methods between classification and generative settings. Specifically, we calculate their Spearman's Rank Correlation Coefficient, which is 0.59, indicating a moderate positive monotonic correlation. The results are detailed in **Appendix C (Figure 11)**.
> - **[Increasing the Classification Threshold Can Better Align the Two Settings]** As indicated in Section 4.4 and Figure 11, "the model tends to exhibit slightly higher refusal rates during classification compared to generation” . To enhance the classification's representative power, we can adjust the classification threshold for determining whether a model refuses a response, increasing it from _0.5_ to _0.7_. This can improve the correlation of DSR between classification and generative settings, raising the correlation coefficient from _0.59_ to _0.64_.
> ***
> > **Q1:** The paper performs the entire evaluation on a single LVLM - LLaVA-1.5. It would be interesting to see if the findings generalize to other LVLMs like CogVLM and InternLM-XC.
> - Thank you for your insightful suggestion. To evaluate the generalizability of the identified mechanisms, we conduct additional analysis on advanced LVLMs, including **LLaVA-Next** (specifically LLaVa-V1.6-Mistral-7B with a different LLMs backbone), **Qwen2-VL** (qwen2-vl-7b with a different training paradigm) and **Pixtral** (specifically pixtral-12b with a different model architecture). The results, presented in **Appendix B.3 (Figure 6,7,8)**, demonstrate that these LVLMs also exhibit the same two mechanisms, and two ensembles strategies generally achieve similar effects as LLaVA-1.5, further supporting the robustness and applicability of our findings.
> ***
> > **Q2:**  The mechanistic analysis characterizes defenses as either safety shift or harmfulness discrimination. However, Figure 3 (b) shows that combination of SR + MO i.e. two safety shift techniques lead to an increase in distance from 0.5 to 0.63. This suggests that defenses might be providing different degrees of both safety shift and harmfulness discrimination.
>
> Thank you for your insightful observation.
> - **[Distance Increase Is Not the Key Difference]** The main difference between these two mechanisms is not whether the distance increases. As noted in Lines 241–242, safety shift primarily enhances the model's general safety awareness, wbroadly increasing refusal tendencies for **both harmful and benign queries**. This may increase their distance **but not necessarily**, as seen with the _Responsible_ method in Figure 2 (also mentioned in Line 80: "without necessarily widening the gap between their refusal distributions"). In contrast, harmfulness discrimination either increases the refusal probabilities for harmful queries, or decreases them for benign queries, or both. This typically results in a more pronounced increase in distance.
> - **[SR + MO’s Distance Increase Stems from Safety Shift]** From Figure 2, SR + MO achieves stronger safety awareness which simultaneously increases refusal probabilities of harmful and benign queries. The observed distance increase to 0.63 occurs because refusal probabilities for harmful queries grow at a much higher rate compared to benign queries. This illustrates that these mechanisms are not strictly mutually exclusive in their impact on distance but may overlap in practice. We will further clarify this in the final version of the paper.

---

> ### Author Response · Authors · 2024-11-26
> **Friendly reminder to respond to author rebuttal**
>
> Dear Reviewer m8ZD,
>
> Thank you again for your review! As you know, we are quickly approaching the draft update deadline Nov 27 EOD.
>
> In light of this, we would love to hear your thoughts about our rebuttal, including whether it sufficiently addresses your concerns and questions, and whether you would like to see further edits made. Any feedback is welcome and greatly appreciated!
>
> Sincerely,
> Authors

---

> > ### Comment · Reviewer_m8ZD · 2024-11-27
> >
> > I thank the authors for their effort. Evaluation on text-only LLMs as well as the analysis of generation quality are valuable additions to the paper. I still have two concerns - (1) a correlation of 0.64 is not enough to show that the results in the generation setting are actually caused by the mechanistic insights in the classification setting, (2) I am not convinced that existing defenses can be cleanly classifier into either Safety Shift or Harmfulness Discrimination.

---

> ### Author Response · Authors · 2024-11-28
>
> Thank you for your feedback and recognition of our supplemented results. Below are our responses to your new concerns.
>
> 1. A correlation of 0.64 is not enough to show that the results in the generation setting are actually caused by the mechanistic insights in the classification setting.
> - Similar to reward models, a binary classification setting can reveal a model's intrinsic preferences. This is because the result space in classification tasks is limited (e.g., "safe" or "unsafe"), allowing the model's choices to clearly reflect its preference. In contrast, the output space in generation tasks is much larger and involves greater randomness, which can obscure the model's preferences. For example, a generative model might initially avoid producing harmful content but display bias in longer outputs due to context changes.
> - This randomness can contribute to the imperfect correlation between generation and classification settings. However, this does not suggest that the two mechanisms are absent in generative settings. Employing appropriate sampling and majority-voting strategies could mitigate these inconsistencies and improve the representational power of binary classifiers in generative contexts.
> ---
> 2. I am not convinced that existing defenses can be cleanly classifier into either Safety Shift or Harmfulness Discrimination.
> -  **Not all defense methods fall neatly into these two types.** Our analysis focuses on refusal probability as a measure of safety preference. Through experimental investigation of 28 common individual defense methods, we identify **two main mechanisms in terms of safety preference** as shared traits among those **effective defense methods**. An effective method generally operates based on one of these two mechanisms, though it may not apply in every case/instance. As previously detailed in the appendix **(Figure 5, Page 19)**, System Reminder and Model Optimization typically operate in safety shift, while most Query Refactoring methods work primarily in harmfulness discrimination.
> -  **We also pointed out in our paper (lines 255-258) that less-effective defense methods like Noise Injection do not fit neatly into either of these two categories.**
>
>
> Thank you once again for improving our work. We look forward to your further feedback.

---

### Author Response · Authors · 2024-11-24
**General Response**

### **Summary of Strengths**
We sincerely thank all reviewers for their insightful reviews and utmost helpful suggestions. We are delighted to see that the reviewers appreciated our work with the following strengths:
1. Interesting idea to reformulate jailbreak defenses. (`m8ZD`, `QcMJ`, `U7of`)
2. Insightful analysis and comprehensive experimental evaluation. (`m8ZD`, `XiPx`, `QcMJ`, `U7of`)
3. Valuable defense mechanisms and effective ensemble strategies. (`XiPx`, `QcMJ`)
4. Well-written and easy to follow. (`m8ZD`, `U7of`)


### **Summary of Updates**
In response to the reviewers’ comments/questions, we have addressed the following items in our rebuttal and updated paper, with updates highlighted in blue. (Supplemented results, primarily in figures or large tables, **are directly included in the PDF**; please refer to them alongside the rebuttal.)
- Demonstrate the **generalizability of our findings on text-only LLMs** `(LLaMA-3.1-8B)` in **Appendix B.4 (Figure 9)**. (m8ZD, XiPx, QcMJ)
- Analyze the **utility/quality of generated response** under various defenses using `MM-Vet dataset`, in **Appendix D (Table 11)**.  (m8ZD, U7of)
- Illustrate the **positive correlation** between classification and generative settings, in **Appendix C (Figure 11)**. (m8ZD, QcMJ)
- Elaborate on the **applicability of our findings to diverse LVLMs** (`LLaVa-V1.6-Mistral-7B`, `Qwen2-VL-7B`, `Pixtral-12B`), in **Appendix B.3 (Figure 6,7,8)**. (m8ZD, XiPx, QcMJ)
- Conduct experiments under **more diverse real-world attack methods** using `JailbreakV-28K dataset`, in **Appendix E (Table 12)**. (XiPx, U7of)
- Summarize and provide **suggestions on ensemble strategies**. (XiPx, QcMJ)
- Clarify the distinction between safety shift and harmfulness discrimination on distance. (m8ZD)
- Discuss the response speed of different defense methods, in **Appendix F (Table 13)**. (XiPx)
- Supplement experiments on DPO methods, in **Table 1 and Table 11**. (U7of)
- Provide more detailed captions for figures and tables. (XiPx)

---

### Meta-Review · Area_Chair_QqyV · 2024-12-24

**Metareview:**

This paper aims to characterize internal jailbreak defenses into two categories: Safety Shift and Harmfulness Discrimination. The authors achieve this by prompting the model with a classification question and analyzing the output distribution for benign and harmful queries. The authors are encouraged to conduct the evaluations on diverse models.
In addition, the analysis doesn't show statistically significant evidence that existing defenses can be cleanly classified into either Safety Shift or Harmfulness Discrimination. It would be important for the authors to draw conclusions with statistical significance to back up the claims.

**Additional Comments On Reviewer Discussion:**

The reviewers are agreed on the final decision.

---

### Decision · Program_Chairs · 2025-01-22

Reject